

# Offspring and adult chemosensory recognition by an amphisbaenian reptile may allow maintaining familiar links in the fossorial environment

José Martín[1], Ernesto Raya-García[2], Jesús Ortega[1] and Pilar López[1]

[1] Departamento de Ecología Evolutiva, Museo Nacional de Ciencias Naturales, CSIC, Madrid, Spain
[2] Laboratorio de Herpetología, Instituto de Investigaciones Sobre los Recursos Naturales, Universidad Michoacana de San Nicolás de Hidalgo, Morelia, Michoacán, Mexico

## ABSTRACT

Kin recognition is a phenomenon with an important function in maintaining cohesive social groups in animals. Several studies have examined parent–offspring recognition in species with direct parental care. Few studies have, however, explored parent–offspring recognition in animals that, at best, only show apparent indirect parental care, such as some reptiles. In this study, we investigated reciprocal parent–offspring recognition in the fossorial amphisbaenian *Trogonophis wiegmanni*, a viviparous species that shows potential stable 'family groups' in the form of parent-offspring long-term associations. We examined whether adult males and females could discriminate via chemical cues between familiar juveniles which associate with them within their family groups, and are potentially their offspring, to that of unfamiliar juveniles, and whether juveniles could discriminate between familiar adult males and females of their family group (probably their parents) and unfamiliar unrelated adults. We measured tongue flick behavior to study chemosensory responses to the scent of conspecifics. We found that adult female amphisbaenians, but not males, could discriminate between scents of familiar and unfamiliar juveniles. Juvenile amphisbaenians did not discriminate between familiar and unfamiliar adult females, but recognize familiar from unfamiliar males. We discuss our results of parent–offspring recognition according to its potential social function in an ecological fossorial context where visibility is limited and chemosensory kin recognition may contribute to the establishment of stable family groups.

# INTRODUCTION

In social species that present parental care or form family groups, the ability to recognise their own offspring or their own parents and siblings (i.e., kin-recognition) is crucial to maintain long-term stable family associations (*Clutton-Brock, 1991*; *Halpin, 1991*; *Tang-Martinez, 2001*). Although social and family aggregations are widespread in many animals, this is not the case in reptiles, which only rarely show parental care or stable parent–offspring association and social groups (reviewed in *Gardner et al., 2016*; *Whiting & While, 2017*; *While et al., 2019*), yet viviparity seems to be an important factor in the evolution of sociality

Corresponding author
José Martín, mcnmr2g@mncn.csic.es

in reptiles (*Halliwell et al., 2017*). Cohesive family groups, frequently stable for several seasons, has been previously seen to occur in a few species of viviparous skinks (*Bull & Baghurst, 1998; Duffield & Bull, 2002; Chapple, 2003; O'Connor & Shine, 2004; Langkilde, O'Connor & Shine, 2007; Gardner et al., 2016*)). An increased understanding of the diversity of social life in reptiles prompts deeper questions about the existence and characteristics of recognition mechanisms by which groups are maintained. For example, mother–offspring and between group member recognition has been found in *Egernia*- clade skinks (*Bull et al., 1994; Main & Bull, 1996; Bull et al., 2000; O'Connor & Shine, 2006*), but also in some viviparous lizard species without parental care, such as common lizards (*Lacerta vivipara*; *Lena & De Fraipont, 1998*). Sibling recognition has been described in juvenile tree skinks *Egernia striolata* (*Bull et al., 2001*) and in hatchling green iguanas (*Werner et al., 1987*). Conspecific- and kin-recognition in lizards is often mainly based on chemical cues (*Bull et al., 2000*; reviewed in *Mason & Parker, 2010; Martín & López, 2011*), although in most of these studies the use of additional cues was not discounted.

Amphisbaenians are a major distinctive group of fossorial reptile (*Gans, 1978; Gans, 2005*), however, there is very limited information on their social behavior and ecology; likely because their underground life provides a host of research challenges (*Henderson et al., 2016*). The amphisbaenian *Trogonophis wiegmanni* is a NW African Mediterranean species found from Morocco to northeast Tunisia (*Bons & Geniez, 1996*). Individuals spend all their lives buried in sandy soils, which lay below leaf litter, and can be commonly found under rocks (*Civantos, Martín & López, 2003; Martín, López & García, 2013*). Interestingly, this amphisbaenian is often found in potential stable 'family groups' (*Martín et al., 2011a; Martín et al., 2011b*), where the same individuals can be relocated together under the same or nearby rocks on different days within the same season (J Martín, 2020, unpublished data). In contrast to the oviparous reproductive system of most amphisbaenian species (*Gans, 1978; Andrade, Nascimento & Abe, 2006*), *T. wiegmanni* is viviparous and bears live young at the end of summer (*Bons & Saint Girons, 1963*). After birth, in early autumn, juveniles are often found in close proximity to a pair of adults or at least one adult individual, usually a female (*Martín et al., 2011a; Martín et al., 2011b*), and successive recaptures suggest that this association is often maintained for several months into the next season (J Martín, 2020, unpublished data). These observations strongly suggest that juveniles might remain with their parents until they are older and, therefore, that some long-term parent–offspring association might occur (*Martín et al., 2011a*). These simple suggested forms of parent–offspring associations are similar to those identified for other viviparous lizard species, including those that live in stable family groups (e.g., *Gardner et al., 2016; Halliwell et al., 2017*). However, the importance of these social aggregations in this amphisbaenian and whether and how social recognition occurs are unknown.

Amphisbaenians have conspicuous morphological and functional adaptations to a fossorial life, such as reduced vision, elongated body and loss of limbs (*Gans, 1974; Gans, 1978; Gans, 2005; Navas et al., 2004*). However, these adaptations constrain many aspects of their ecology. The ecological demands of the fossorial underground environment and the responses of amphisbaenians are often very different from those of terrestrial epigeal reptiles that live over the ground surface (e.g., *Papenfuss, 1982; Martín, López & Salvador, 1990;*

*Martín, López & Salvador, 1991*; *Colli & Zamboni, 1999*; *Webb et al., 2000*). The detection and identification of conspecifics is one of the major ecological problems of the fossorial environment. Amphisbaenians have only rudimentary vision and the utility of visual cues is clearly limited underground (*Gans, 1978*). Thus, chemoreception may play an important function in detecting and identifying conspecifics, as it occurs in many squamate reptiles (*Mason & Parker, 2010*; *Martín & López, 2011*). The amphisbaenian *Blanus cinereus* uses chemical cues in conspecific and sex discrimination and in self-recognition (*Cooper, López & Salvador, 1994*; *López, Cooper & Salvador, 1997*; *López & Martín, 2009*), and this species is thought to be able to scent-mark and identify its own home range (*López, Martín & Barbosa, 2000*). Similarly, both male and female adult *T. wiegmanni* can discriminate the scent of an adult individual with which they had formed a pair bond from an unfamiliar individual of the same sex as the partner, and males, but not females, have self-recognition abilities (*Martín et al., 2020*). We hypothesized that similar chemosensory abilities could allow the unexplored possibility of kin-recognition, and, therefore, that conspecific chemical cues may be very important in the formation and maintenance of stable family groups in fossorial animals.

In this paper, we tested the ability of *T. wiegmanni* amphisbaenians to detect and discriminate by using chemical cues alone between familiar (likely related) individuals that were found together forming potential stable family groups (i.e., kin-recognition) and unfamiliar (likely unrelated) individuals. We specifically examined: (a) whether adult amphisbaenians were able to recognize the juveniles that were found in their social 'family' group, which we are assuming are their offspring, against other unfamiliar juveniles, and (b) whether juvenile amphisbaenians were able to recognize and discriminate between adult males and females, and to discriminate between the adults found in their social groups, which very likely could be their parents, and other unfamiliar adult individuals. We discuss how chemosensory discrimination of conspecifics may contribute to the formation of social relationships in fossorial animals.

## MATERIALS AND METHODS

### Study site and study animals

We conducted field work at the Chafarinas Islands (Spain) during April. This is a small volcanic archipelago located in the southwestern area of the Mediterranean Sea (35°11′N, 2°25′W), 4.6 km off of the northern Moroccan coast (Ras el Ma, Morocco) (*Martín et al., 2011b*; *Martín et al., 2011c*). The climate is Mediterranean, dry and warm, and vegetation consists of bushes adapted to salinity and drought (Genus *Suaeda*, *Salsola*, *Lycium* and *Atriplex*). Populations of the amphisbaenian *T. wiegmanni* are very large in these islands (*Martín et al., 2011c*).

We followed different routes between 07:00 and 18:00 (GMT) and lifted most rocks found as amphisbaenians were found active under these rocks (*López, Civantos & Martín, 2002*). When we found a possible familiar group of amphisbaenians (i.e., two adults, male and female, and one juvenile), we captured all individuals by hand. We used a metallic ruler to measure snout-to-vent length (SVL, adults: mean $\pm$ SE = 148 $\pm$ 4 mm; juveniles:

mean ± SE = 85 ± 2 mm) We examined cloacas carefully and everted the hemipenes of males to determine sexes of adults. In all cases that two adults were found together under the same rock they were a male and a female. Juveniles, according to their body size, were individuals born at the end of the previous summer (see *Bons & Saint Girons, 1963*; *Martín et al., 2011b*; *Martín et al., 2012*). Juveniles could not be sexed with reliability. Groups made up 24% of records, and single adults were not collected.

We followed recommended procedures for the transport of live reptiles (*ASIH, 2004*) to transport amphisbaenians to the laboratory. Family groups were kept together in separate plastic boxes with sand from the capture area. The same day after starting the journey, we housed amphisbaenians at "El Ventorrillo" Field Station (Navacerrada, central Spain). Individuals found in a group in the field were kept together in the same indoor plastic terrarium ($40 \times 30 \times 30$ cm), one for each group, throughout the whole experiment. Each terrarium had a loose coconut fiber substrate (5 cm depth) and a flat tile ($20 \times 20$ cm) on the substrate surface to allow amphisbaenians to forage and thermoregulate under it (*López, Salvador & Martín, 1998*; *López, Civantos & Martín, 2002*). Amphisbaenians could attain an optimal body temperature by thigmothermy with the substrate warmed by a heating cable placed below the terraria, connected to a thermostat (*Gatten & McClung, 1981*; *López, Civantos & Martín, 2002*). The room was only illuminated with natural sunlight entering through large windows, so that the photoperiod was that of the region, although amphisbaenians spent all the time buried underground. We fed amphisbaenians three times per week mealworm larvae and pupae, snails and freshly pre-killed crickets, dusted with a multivitamin powder (*Goetz, 2005*; *Martín et al., 2013b*). We placed these prey under the tiles where amphisbaenians readily ate within a few hours. We moistened the substrate with a water spray frequently to avoid desiccation and to provide drinking water. All the individual amphisbaenians were healthy and monthly checks showed that they maintained or increased their original body mass.

Field study and capture of amphisbaenians were approved by the Spanish "Dirección General de Calidad y Evaluación Ambiental y Medio Natural" of the "Ministerio de Agricultura, Alimentación y Medio Ambiente" (number 12706). Research procedures were approved by the "Comisión Ética de Experimentación Animal (CEEA)" of the Museo Nacional de Ciencias Naturales, CSIC.

## Chemosensory tests

In June, we designed an experiment in the laboratory to estimate detection and discrimination of conspecific chemical cues by *T. wiegmanni* amphisbaenians. For this, we used measures of tongue-flick (TF) behavior in response to chemical stimuli presented on cotton swabs. This swab test provides a rapid and reliable bioassay of the ability of reptiles to respond differentially to biologically relevant scent stimuli (*Cooper & Burghardt, 1990*; *Cooper, 1994*; *Cooper, 1998*). This is based on that tongue-flicking behavior functions to sample chemicals for vomerolfactory analysis (*Halpern, 1992*). The existence of a correlation between elevated TF rates and vomeronasal organ use, and the necessity of an intact vomeronasal system for normal TF responses to scents have been experimentally tested (*Graves & Halpern, 1990*; *Halpern, 1992*). It is assumed that an increase in TF rates

in response to a scent stimulus, with respect to the basal TF rates, indicates detection of that scent, and that differential TF rates to different chemical stimuli indicate discrimination of the different stimuli (*Cooper & Burghardt, 1990*; *Cooper, 1994*; *Cooper, 1998*; *Martín et al., 2020*). We compared TF rates of amphisbaenians in response to scents of different types of conspecifics and we also measured responses to distilled water as an inodorous scent control to gauge baseline TF rates in the experimental setup (*Cooper & Burghardt, 1990*).

Because the groups of amphisbaenians used in this study were found at well separated field sites (more than 50 m between the nearest locations) and amphisbaenians have a low dispersal ability underground (J Martín, 2020, unpublished data), we assumed that individuals of each group had not had previous contact with individuals from other groups and were considered as unfamiliar individuals, whereas individuals within each group were considered as familiar individuals.

We designed a first experiment to test whether adult amphisbaenians display juvenile recognition (i.e., offspring recognition). We tested the responses of adult male ($n = 10$) and female ($n = 10$) amphisbaenians to (a) water (control), and scents of (b) an unfamiliar juvenile that had never been in contact with the responding amphisbaenian and (c) the familiar juvenile that was originally found in the field with the responding adult amphisbaenian (i.e., its potential offspring) and that shared its terrarium during all the study.

In a second experiment, we tested for sex and familiar recognition of adults by juvenile amphisbaenians (i.e., discrimination of adult males and females and recognition of adult members of their groups that might be their potential parents). We examined the responses of juveniles ($n = 14$) to (a) water (control), and scents of four classes of adult amphisbaenians: (b) the familiar male and (c) the familiar female that were originally found in the field together with the juvenile and that shared its terrarium during the study, and (d) an unfamiliar male and (e) an unfamiliar female that had never been in contact with the juvenile.

To prepare the scent stimuli, we dipped the cotton tip (1 cm) of a wooden applicator (10 cm) in deionized water and then rolled the moistened cotton over the cloaca of the donor amphisbaenian. Each individual amphisbaenian was used as donor of scent in two occasions in different days (as familiar or unfamiliar individual). Because, we tested in two different experiments the responses of adult to juvenile scent and *vice vers* a, donor individuals had enough time to recover from any possible perturbation of this handling before being used as responding individuals.

Before starting the experiment, we gently took the responding amphisbaenian from its home terrarium and placed it in a clean testing cage ($20 \times 15$ cm) that contained a very shallow and loose clean substrate of coconut fiber (0.5 cm depth). We left the animal there for 15 min for acclimation to the new cage before undertaken the tests. This procedure allowed us to observe the amphisbaenian responses while they were semi-buried and behaving normally, as when they were completely buried in their terraria (i.e., without showing any signs of stress such as rapid escape locomotion or defensive behavior typically observed when amphisbaenians were brought above the surface). We made observations under a red light in the partially darkened laboratory to avoid disturbing amphisbaenians.

Since chemoreceptive responses in reptiles can depend on temperature (*Van Damme et al., 1990*), we maintained air temperature in the laboratory at 22 °C, close to the preferred temperature of *T. wiegmanni* (*Gatten & McClung, 1981*; *López, Civantos & Martín, 2002*). All animals had been acclimatized to laboratory conditions and experimenter's presence for at least two months before testing.

In each trial, the same experimenter (PL) in all cases, who was blind to the treatments, slowly moved the cotton swab to a position 2 cm anterior to the snout of the amphisbaenian, previously placed within the testing cage, and recorded the number of TF directed to the swab for 60 s beginning with the first TF. Amphisbaenians responded to the scent stimuli and directed TFs to the swab in all treatments and tests. We predicted that, if amphisbaenians had the ability to detect conspecifics using chemosensory communication, then TF rates to conspecific scents should be greater than to the control water. Differential TF rates to the different categories of conspecifics would indicate discrimination of these categories.

Each individual was tested with one single stimulus per day, and in subsequent days it was tested with the rest of treatments. Each cotton swab with a scent stimulus was used once in a single test of an individual, and then thrown away and a new one used for the next focal animal. Order of presentation of the stimuli was randomized. We conducted trials between 1100 h and 1500 h (GMT) when the amphisbaenians were fully active. After the tests, we immediately returned each amphisbaenian to its home terrarium. To remove any chemical from the used testing cage, we thoroughly rinsed it with clean water and soap and left it to dry outdoor before being used in another test with a new coconut fiber substrate. We used several identical cages for different tests.

## Data analyses

To test for differential chemosensory responses of amphisbaenians to the different chemical conspecific stimuli, we used repeated measures General Lineal Models (GLMs) to test for differences in the numbers of TFs (dependent variable) among 'treatments' (water and the different conspecific scents) as a repeated measures factor. In the first experiment, we also included in the model the sex of the responding adult amphisbaenian as a fixed factor, and the interaction of sex with treatment. Residuals of the models fulfilled the normality and homoscedasticity assumptions after log-transformation of the numbers of TFs (tested with Shapiro–Wilk's and Hartley's $F$ max tests). We used post-hoc Tukey's tests for comparisons of TF rates among treatments to test for discrimination of control water and the different conspecific chemical stimuli.

## RESULTS

### Juvenile recognition by adult amphisbaenians

There were significant differences in TF rates of adult amphisbaenians among scent treatments (repeated measures GLM, $F_{2,36} = 52.70$, $p < 0.0001$) but there were no significant differences in overall TF rates between sexes ($F_{1,18} = 0.23$, $p = 0.64$) and the interaction between treatment and sex was significant ($F_{2,36} = 5.47$, $p = 0.0084$; Fig. 1).

Both males and females had significantly higher TF rates to cotton swabs bearing any of the juvenile stimuli than to the blank cotton swabs with water (Tukey's tests, $p < 0.0003$

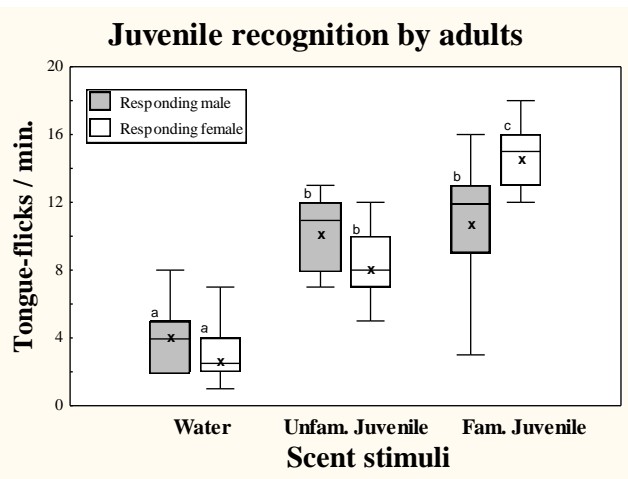

**Figure 1** **Responses of adult *T. wiegmanni* amphisbaenians to juvenile scent.** Box-whiskers plots for the number of directed tongue-flicks emitted by male (hatched boxes) and female (open boxes) adult amphisbaenians in 60 s in response to control water or chemical stimuli of familiar or unfamiliar juveniles presented on cotton swabs. Letters indicate statistically significant differences in post-hoc Tukey's tests between treatments.

in all cases). In males, there were no significant differences between TF rates to the scent of the familiar or an unfamiliar juvenile ($p = 0.99$), while in females, the scent of familiar juveniles elicited higher TF rates than the scent of unfamiliar juveniles ($p < 0.05$; Fig. 1).

Additionally, the range of TFs responses to scent of familiar juveniles was greater in males than in females (Fig. 1), and while all individual females showed higher responses to the familiar than to the unfamiliar juvenile, only 50% of males showed higher responses to the familiar juvenile.

### Responses of juvenile amphisbaenians to adult scent

There were significant differences in TF rates of juvenile amphisbaenians among scent treatments (repeated measures GLM, $F_{4,52} = 24.49$, $p < 0.0001$; Fig. 2). Post-hoc tests showed that TF rates of juveniles to water were significantly lower than to any conspecific stimuli (Tukey's tests, $p < 0.0006$ in all cases), showing that all conspecific scents were detected. Responses to scent of an unfamiliar male were significantly lower than to the familiar male ($p < 0.03$) and familiar ($p = 0.0036$) or unfamiliar female ($p = 0.0085$). The TF rates to the familiar male, however, did not significantly differ of TF rates to the familiar ($p = 0.95$) or unfamiliar female ($p = 0.99$), which did not differ ($p = 0.99$; Fig. 2).

### DISCUSSION

Our results are consistent with the hypothesis that *T. wiegmanni* amphisbaenians are able to detect and to discriminate among several categories of conspecifics using chemical cues alone. Specifically, the results of this study strongly suggest that adult female amphisbaenians are capable of, presumed, offspring recognition. Moreover, juvenile amphisbaenians are able to partly recognize the adults with which they have been

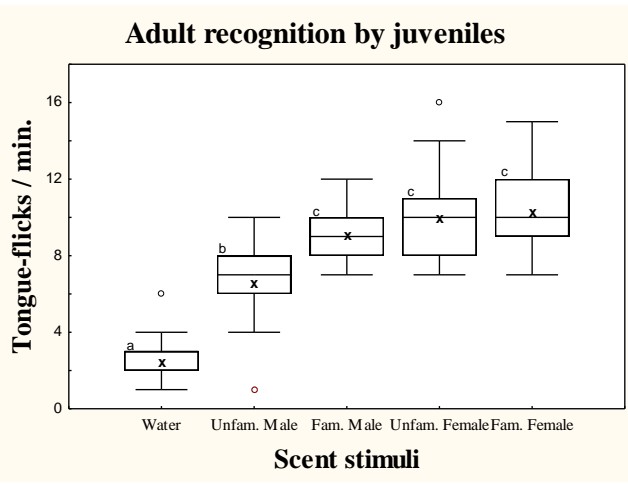

**Figure 2 Responses of juvenile *T. wiegmanni* amphisbaenians to adult scent.** Box-whiskers plots for the number of directed tongue-flicks emitted by juvenile amphisbaenians in 60 s in response to control water or chemical stimuli of familiar or unfamiliar adults presented on cotton swabs. Letters indicate statistically significant differences in post-hoc Tukey's tests between treatments.

associated in the long-term, which probably are their parents. However, adult males did not discriminate between familiar and unfamiliar juveniles, and juveniles did not discriminate between familiar and unfamiliar adult females.

With respect to the discrimination of scents of juveniles by adult *T. wiegmanni* amphisbaenians, there were clear intersexual differences; adult males detected and discriminated the juvenile scent from water, but these males did not discriminate between an unfamiliar juvenile and the familiar juvenile that was found with the male in the field and that shared his terrarium in the experimental situation. This result is very interesting because in many animal species, males through distinct sensory pathways can discriminate their own offspring (familiar) from non-related offspring (unfamiliar) and with this strategy they mediate protective or infanticide behaviors (*Waldman, 1988*; *Elwood, 1991*; *Elwood, 1992*; *While, Uller & Wapstra, 2009*). It is likely that male *T. wiegmanni* may be able to discriminate the scent of familiar juveniles from adult individuals, but there might not be an adaptive function to discriminate their offspring if there are no significant benefits for their fitness (e.g., *Beecher, 1991*; *Kempenaers & Sheldon, 1996*). Likely adult male *T. wiegmanni* would not commit infanticide for this reason, but neither offer protection to their offspring. Nevertheless, alternatively, it could be possible that there was a confounding statistical effect when considering the response of all males on average due to the mixed chemosensory responses of different individual males (i.e., some individuals but not others elicited higher responses to familiar than to unfamiliar juveniles). These results might lead to speculate that the responses were different because only some individual males, but not others, were actually the fathers of the familiar juvenile tested, and, thus, perhaps males might be able to recognize their offspring. More experiments where the actual paternity relationships were known are clearly needed to test this hypothesis.

In contrast, adult females discriminated and showed higher chemosensory responses to scents of the familiar juvenile in comparison with an unfamiliar juvenile. Offspring chemical recognition by females has been reported in some lizard species (*Main & Bull, 1996*; *O'Connor & Shine, 2006*; *Head et al., 2008*). This recognition in animals without direct parental care may be important to avoid interference competition such as reducing parent aggression and seeking to establish territories near their kin (*Bull & Baghurst, 1998*; *Lena & De Fraipont, 1998*; *O'Connor & Shine, 2004*; *Head et al., 2008*). Juveniles may benefit from staying with their mothers; for example, in some skinks, females show high levels of conspecific aggression during the postpartum period that may avoid infanticide by other individuals, such that offspring from more aggressive females have higher survival (*O'Connor & Shine, 2004*; *Sinn, While & Wapstra, 2008*). These simple recognition mechanisms may be the first evolutionary steps towards more complex forms of parental care and more complex forms of family life.

These intersexual differences in offspring recognition might be explained by the different probabilities of genetic relatedness between juveniles and the adult male or female found together, which could be linked to the viviparous reproductive system of this amphisbaenian (*Bons & Saint Girons, 1963*). Thus, it seems very likely that the juvenile found associated with a female was her offspring, since the association could have begun with the birth of the juvenile. A female might, therefore, suffer a selective pressure to recognize her offspring from other juveniles, especially if females provided some kind of parental care or protection to their offspring (*Neff, 2003*). In contrast, a male found close to a juvenile, may or may not be the actual father since mating occurs in spring and juveniles were born at the end of summer and the rates of multiple paternities and the potential for factors like long-term sperm storage remain unknown for this species. In this case, there is the potential that males do not provide active protection to juveniles, even if they were found together. Therefore, males would not be selected to recognize their offspring, even if the familiar (or actually related) juvenile was found often in contact with the male. In fact, although we focused this study on groups formed by a male, a female, and a juvenile, it is more frequent to find in the field a female alone with a juvenile (*Martín et al., 2011a*).

Juvenile *T. wiegmanni* amphisbaenians clearly detected scents of adults with respect to the blank control (water) and showed higher chemosensory responses to either the familiar or an unfamiliar female, but also to the familiar male, while the scent of an unfamiliar male was also detected but received lower responses. Recognition by association occurs when an animal learns the particular distinctive signals of familiar individuals around it and perceive these as kin (*O'Connor & Shine, 2006*). If juveniles received some benefits from being associated to their parents (or just to the adults in the group, even if they were not their actual parents), juveniles should be able to detect and discriminate their scent in order to identify and follow them in the underground environment. This behavior would confer substantial fitness advantages in juveniles allowing them to share the territory and other resources of familiar and experienced adults in an environment where resources are perhaps limited.

The lack of discrimination of some categories of adults may reflect that juveniles show a generalized response to any nearby conspecific adult, which might suggest that some
learning or previous experience with different adult individuals is required for a more accurate individual identification (*Tang-Martinez, 2001*; *Frommen, Luz & Bakker, 2007*). This may be explained because juveniles, which have very limited movement rates, would only rarely find adults other than those in their respective social groups. Incomplete experience (recognition by association) with the scent of adult individuals might explain the observed higher responses to any female, similar to those to the familiar male, and the lowest responses to the unfamiliar male. Juveniles might not have a selective pressure to discriminate between individual females because, as they are viviparous, offspring are always going to be associated with their mother upon birth. Also, it is possible that there is low interindividual variation in the chemical scent of females which inhibits individual recognition of particular females. In contrast, chemical differences between the scent of different males might be higher, providing an easier identification of the familiar male. Similarly, in some lizards and other amphisbaenians, males have a higher number, diversity and interindividual variability of lipophilic compounds in femoral or precloacal secretions than females (*García-Roa et al., 2016*; Martín & López, 2006). Nevertheless, in *T. wiegmanni*, both male and female adults are capable of discriminating between cloacal scent of familiar and unfamiliar partners (*Martín et al., 2020*), suggesting that interindividual chemical signatures are different enough as to allow discrimination, at least after some learning. There may be stronger selection for juveniles to recognize different individual males because the male found in a group might or might not be the father of the juvenile. Moreover, if adult males were capable of discriminating between their offspring and other juveniles, adult males might be more territorial and aggressive towards unrelated juveniles, and, then, these juveniles should be able to detect and avoid unfamiliar males.

Many studies across a host of different species of lizards and snakes have used tongue-flick behavior as an indirect way of measuring vomerolfaction (*Cooper & Burghardt, 1990*; *Halpern, 1992*; *Cooper, 1994*; *Cooper, 1998*). These studies assume that differences in TF rates between stimuli indicate discrimination of a scent, but also that identification may occur with only a few TFs, being a further increase in TFs a reflect of a 'higher interest' for a given scent. Therefore, the differences in TF rates of *T. wiegmanni* observed in our experiment can confidently be considered as chemosensory recognition and discrimination of different scents. The lack of differences observed between some stimuli, however, indicated that both scents elicited a similar interest, although not necessarily in all cases, that they were not recognized as different scents. Alternatively, the cloacal chemical cues used in our study might not provide complete information and amphisbaenians may need additional cues, or additional learning, to achieve a complete identification of conspecifics. If this is the case, further experiments examining other stimuli and other behavioral responses may be needed to determine the extent of the conspecific discrimination abilities of amphisbaenians.

Although we lack data of the actual paternity and genetic relatedness between the adults and the juvenile amphisbaenians of each group that we found together in the field, our results may suggest that kin-recognition occurs in at least adult female *T. wiegmanni*. The conditions of the fossorial environment may explain the observed low dispersal ability and high site fidelity of individual *T. wiegmanni* amphisbaenians (J Martín, 2020, unpublished

data). Therefore, it is very likely that, because of the viviparous reproduction, the juvenile and at least the female found together were genetically related. Further studies should consider a design to compare the responses of truly genetically related adults and juveniles with those of individuals that are unrelated but familiar because they are experimentally placed together. Indeed, such an experimental design would also reveal the mechanism for which recognition of related individuals can occur. Kin-recognition may occur by 'association', when animals learn to recognize the individual signals of other animals that live together, and that are thereafter considered as kin, or by 'phenotype matching', when animals use a reference phenotype (either self or kin) against which other individuals are compared (*Halpin, 1991*; *Tang-Martinez, 2001*). In lizards, both mechanisms have been found (*Bull et al., 2001*; *O'Connor & Shine, 2006*), yet for fossorial amphisbaenians, with low mobility, it might be more likely that recognition by association is occurring, if the probability of finding unrelated individuals in the group is low. This hypothesis remains to be tested and future studies are needed.

## CONCLUSIONS

Our findings suggest that chemosensory kin-recognition may allow amphisbaenians to recognize offspring and tolerate relatives or familiar individuals, to maintain stable family associations of, at least, mother and offspring, to reduce kin competition and intraspecific aggression. Our findings give insight into an early stage in the evolution of kin-recognition and the establishment of familiar groups as they suggest that individual amphisbaenians may benefit from being capable of chemosensory recognition of conspecifics. There could be initial selection for chemosensory detection of familiar individuals, but animals at an early stage of familiar grouping may not have evolved the means to assess the relatedness of conspecifics. Only as the system becomes more efficient might detection of related individuals benefits accrue, and selection might operate on individuals to perceive this and to maintain long-term family groups. Future studies should examine the mechanisms for which recognition occurs, how social associations are maintained and the benefits of these social aggregations for juvenile and adult amphisbaenians in the fossorial environment.

## ACKNOWLEDGEMENTS

We thank G Alexander, K Strickland, and an anonymous reviewer for their helpful comments. We also thank the field station of the "Refugio Nacional de Caza de las Islas Chafarinas" for logistical support and "El Ventorrillo" MNCN Field Station for use of their facilities. We thank JI. Montoya, J Díaz, G Martínez, A Sanz, F López, A Ruiz, and J Zapata for support and friendship in the Islands.

### Funding

Financial support was provided by a contract from the Organismo Autónomo de Parques Nacionales (Spain) and by the Ministerio de Ciencia, Innovación y Universidades project

PGC2018-093592-B-I00 (MCIU/AEI/FEDER, UE). The publication fee was paid by the CSIC Open Access Publication Support Initiative through its Unit of Information Resources for Research (URICI). There was no additional external funding received for this study. The funders had no role in study design, data collection and analysis, decision to publish, or preparation of the manuscript.

## Grant Disclosures

The following grant information was disclosed by the authors:

Organismo Autónomo de Parques Nacionales (Spain).

Ministerio de Ciencia, Innovación y Universidades: PGC2018-093592-B-I00 (MCIU/AEI/FEDER, UE).

CSIC Open Access Publication Support Initiative through its Unit of Information Resources for Research (URICI).

## Competing Interests

The authors declare there are no competing interests.

## Author Contributions

- José Martín conceived and designed the experiments, performed the experiments, analyzed the data, prepared figures and/or tables, authored or reviewed drafts of the paper, and approved the final draft.
- Ernesto Raya-García performed the experiments, authored or reviewed drafts of the paper, and approved the final draft.
- Jesús Ortega and Pilar López conceived and designed the experiments, performed the experiments, authored or reviewed drafts of the paper, and approved the final draft.

## Animal Ethics

The following information was supplied relating to ethical approvals (i.e., approving body and any reference numbers):

Research procedures were approved by the "Comisión Ética de Experimentación Animal (CEEA)" of the Museo Nacional de Ciencias Naturales, CSIC.

## Field Study Permissions

The following information was supplied relating to field study approvals (i.e., approving body and any reference numbers):

Field study and capture of amphisbaenians were approved by the Spanish Dirección General de Calidad y Evaluación Ambiental y Medio Natural of the Ministerio de Agricultura, Alimentación y Medio Ambiente (number 12706).

## Data Availability

The data are available at figshare:

Martin, Jose; García, Ernesto Raya; Ortega, Jesús; López, Pilar (2020): Kin recognition in Trogonophis. figshare. Dataset. https://doi.org/10.6084/m9.figshare.13148033.v1.

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
