# Peer review of "Offspring and adult chemosensory recognition by an amphisbaenian reptile may allow maintaining familiar links in the fossorial environment"

_PeerJ, doi:10.7717/peerj.10780_

## Round 0.1 · original submission · Minor Revisions

This is a fascinating study, and certainly it will greatly contribute to our understanding of reptile social behavior and kin-based sociality – especially in the context of a taxa that is vastly understudied in this regard. This is quite commendable, and the authors should be very pleased. All three reviewers, as well as myself, noted the value and merit present within this study.

The major request from the reviewers is to increase clarity with regards to: 1) your statistical analysis, 2) the study system, and 3) your experimental methods. This should not be too difficult, but would go a long way in providing your future readers the ability to better understand all the components of this study, and thus your work. Each reviewer had different regions of the paper where they felt increased clarity would help, and I would suggest tackling each one to form a fully flushed out manuscript.

The reviews of this manuscript also - and staying in the same vein as ‘increased clarity’ - have noted that it could use some more polishing with respect to writing and structure. This a generally small hurdle but an important one, and I suggest the authors take care to pay close attention to this while they are revising the manuscript. The reviewers have kindly provided a combined, extensive list of suggestions to improve the readability of this manuscript which should go a long way in helping the authors bring this paper to a top-notch and publishable state. Professional science writing can be challenging, and many of us – including myself – can benefit from having other colleagues weigh in. I know I certainly was not an English major during my university schooling, and I regularly depend on friends and colleagues to lend another set of eyes to improve my scientific writing style and format when I am finalizing a manuscript. I am definitely not saying “find a native English speaker”, because that is a ridiculous request. In my experience, most native English speakers need help with professional writing style and structure, just like everyone else. Rather, I am suggesting having a colleague skilled in scientific writing, who may not be too familiar with your study or the system, read over it to make suggestions about clarity and ease of reading and understanding.

In short, this is a neat study and has all the making for a great paper. With some work to increase clarity and readability, I think it will result in a wonderful publication. I am looking forward to reading your resubmission once you are able to address these, and all of the reviewer’s, suggestions and comments.

·

Basic reporting

The manuscript does need some polishing. There are several instances where inappropriate words are used, and sentence structure is clumsy. The comments below are aimed at improving the manuscript in this regard.
There is a bit of a gap in the literature that you cite. I would include some more recent overviews such as:
While, G. M., M. G. Gardner, D. G. Chapple, and M. J. Whiting. 2019. Stable social grouping in lizards. Behavior of Lizards:321–339.
Whiting, M. J., and G. M. While. 2017. Sociality in lizards. In D. R. Rubenstein, P. Abbot (Eds.) Comparative social evolution (pp. 390–426), Cambridge University Press. Cambridge, UK.
L20: Change ‘cohesive social role in many animals’ to ‘function in maintaining cohesive social groups in animals’.
L21: Change ‘with an evident direct parental care’ to ‘with direct parental care’.
L61: Change ‘that difficult these studies’ to ‘that make these studies difficult’.
L63: This goes from singular (the species) to plural (They). I would recommend starting this sentence as ‘Individuals spend (not spent as is currently written) all their lives buried…’.
L67: Change ‘in’ to ‘on’.
L70: Change ‘mainly’ to ‘usually’.
L71: Change ‘during’ to ‘for’.
L72: Change ‘suggests’ to ‘suggest’.
L77: Change ‘or’ to ‘and’ and drop the ‘)’.
L80: I would change ‘epigeal’ to ‘terrestrial’.
L84: I would change ‘a main role’ to ‘an important function’.
L88: Singular to plural again.
L115: Change ‘We lifting stones’ to ‘We lifted stones’.
L123: I would change to ‘Groups made up 24% of records, and single adults were not collected.’
L133: I would change to ‘…on the substrate surface…’.
L142: Change ‘in’ to ‘within’.
L144: How often were individuals weighed?
L164: Change ‘able’ to ‘capable’.
L203: What do you mean by ‘counterbalanced’? Do you not mean ‘randomized’?
L237: Delete ‘than to the’.
L243: I would word this to indicate that the individuals are able to detect and discriminate – not the species.
L245: Change ‘able’ to ‘capable’.
L246: Change ‘a juvenile amphisbaenians is able to’ to ‘juvenile amphisbaenians are able to’.
L247: Change ‘with which the juvenile has been long-term grouped’ to ‘with which they have been associated in the long-term’.
L267: Change ‘like’ to ‘such as’.
L269: ffor
L301: Change ‘found’ to ‘find’.
L302: Insert ‘other’ at the start of this line.
L308: Change ‘able’ to ‘capable’.
L310: Change ‘enough different as’ to ‘different enough’.
L312: How could males be more aggressive to unfamiliar juveniles when they cannot discriminate between juveniles?
L323: Change ‘allow to determine’ to ‘reveal’.

Experimental design

Overall, the experiment design is fine and is clearly explained. The statistical analysis appears sound.
The finding that males do not discriminate between associated and non-associated juveniles is very interesting. I wonder if indeed (as you speculate in the discussion) that this may be because there is a lower probability that the male found with the female and juvenile is actually the father. If that is the case, it may be worth checking if there is a greater range in TFs between the males in comparison to the adult females. If, for example, 50% of the males were they actual fathers, those individuals may be expected to have higher TFs (if indeed fathers can recognise their offspring). It should be a very simple thing to check and if that is the case, it would make this a much more powerful study.

Validity of the findings

I think that the manuscript would be strengthened by addition of a paragraph the explicitly considers the methods used. Are differences in TF a reliable way of demonstrating recognition? For example, if a juvenile TFs had similar rates in response to the scent of two adults, one of which is the parent, does this definitely mean that the juvenile cannot distinguish between parent and non-parent? What if the juvenile TFs simply because it is trying to discriminate? And so TF rate may be also impacted by the interest of the animal to the scent presented.

Additional comments

The study reported in this manuscript tests for individual discrimination in a species of amphisbaenian. Family groups were collected from the field and tongue flick rates were used to measure recognition between adults and juveniles. Adult females appear to be able to discriminate between familiar and unfamiliar young. Juveniles appear capable of discriminating between familiar and unfamiliar adult males, but not females. It is really interesting that adult males could not discern the difference between juveniles and that juveniles could not discern between females. Although some possible explanations are provided for the surprising results, they are not entirely convincing. Overall, this manuscript reports some very interesting findings and I would like to see it published. I do think it could do with some polishing and the addition of an extra paragraph in the discussion that explicitly assesses the method (Validity of findings). I think that it would also be worth assessing the TF rates for the adult males to see if there is any pattern that could show if not all males were the actual fathers of the familiar juveniles (see experimental design).

·

Basic reporting

Overall, the manuscript is well written, with results that follow clear hypotheses and with a wide appreciation of literature. However, I noticed in a number of places some grammatical errors. I have tried to highlight these in my general comments that hopefully help to streamline the manuscript.

Experimental design

The research question is clearly defined, and the majority of the experimental design appears to be valid. However, the methods section lacks some detail, rationale and justification regarding some of the more critical decisions about how to collect data and subsequently handle and analyse it. As such, assessing the results is somewhat challenging given the lack of clarity. I have pointed to these in my general comments to the authors, most of which I think can probably be addressed with more information, but without which it is hard to judge whether the results are valid.

Validity of the findings

No comment.

Additional comments

In this manuscript, authors tested whether individual amphisbaenians were able to discriminate between familiar and unfamiliar individuals using chemosensory mechanisms. They conclude that there is some evidence for this, but this depends on the sex of the reciever and provider of the signal. Overall, I really enjoyed reading the manuscript, and learning about the species of which I knew nothing! My minor comments largely refer to some writing hickups throughout the ms. But, I do have a few broader comments which I think the authors should address to help the reader to interpret their results.

1. My major comment is about the measure of tongue flicks as a way of determining discrimination. I am unsure about the validity of this measure, and would suggest that the authors explain the rationale for this much more. My concern here is whether authors are actually measuring recognition, or at least the extent to which the measure reflects discrimination.

2. My other concern lies with the analysis. The section is very brief, and as such it is somewhat difficult to interpret the results. This is because it is unclear (i) how the data were treated, and (ii) how the models were constructed, (iii) how model fit was assessed. Though I think that the authors provide interesting results, it is hard to assess the reliability of these without these details.

3. As a less serious one, I would have liked more information on the species and their social organisation. It is an unfamiliar system, and I would have liked more detail to help to interpret the data and results.


line 60-61; this sentence is not clear, can you edit to clarify your meaning?
line 67; could you expand on this point a bit? For instance, can you elaborate on what sort of "area" are they relocated in, and over what temporal period? It would help to have a more detailed description of their social organisation/social structure
line 77; remove parentheses
line 79-80; to make this more accessible to broader readers, perhaps describe what "fossorial" and "epigeal" mean
line 89; do you mean to mark and identify their own home range?
Line 115; unclear what this searching method refers to. Could you please clarify?
Line 117; What is a "presumably familiar group"? Could you explain what criteria were used to determine this, and why. Also, I assume that the sample size in parantheses here is the number of groups eventually found. It might be better placed later in the manuscript
Line 121; Could you elaborate on what the SVL cut off is that you used to determine juveniles, and how this SVL was determined?
Line 125; I think that having more detailed information about the social organisation of the species earlier could be helpful in interpreting methods here. For instance, it is hard to determine if capturing only individuals that are living in groups is representative
Line 127; Edit to "..were kept together.." or something similar
Line 129 - 132; I think if I understood correctly, individuals found in a group in the field were kept together in the same terrarium. This sentence could be clearer I think.
Line 154; This seems to me a critical assumption in the study, and it would be beneficial to elaborate on the idea that rate of tongue flicks represent detection and discrimination - with discrimination being the key here.
Line 164; should read "..were capable of.."
Line 195; how were tongue flicks determined, and were they recorded as "instances" - so you count the number of TF per trial?
Line 213; what is a within factor?
Line 210; the data analysis section needs a bit more detail. It is not clear what the dependent variable is. I think it might be counts of TF which has then been log transformed. In which case you should really run this model with the raw counts and a Poisson distribution. Either way, I would suggest that the authors provide more detail about the analysis as it is hard to assess the validity of the model structure from the information that is provided.
Line 234 - 240; This is a long sentence with a lot of information and results in, which is somewhat hard to digest without looking at the figure itself. Consider revising to improve clarity.
Line 244; remove full stop
Line 245; here and elsewhere in the manuscript (I noticed quite a few places), "are able of offspring recognition" should be changed to "are capable of ..." or, "are able to recognise offspring"
Line 246; should be either "a juvenile is able to....", or "...juveniles are able to"
Line 269; "ffor" - remove f
Discussion in general; The authors provide some nice detail that could explain the slightly contradictory patterns in the results. However, I think it misses some explicit mentioning of this. For instance, authors conclude that juveniles are able to discriminate between familiar and unfamiliar males, but not females. But on the other hand, conclude that females are able to discriminate between familiar and unfamiliar juveniles. This is a somewhat unexpected pattern, and I was looking forward to the discussion explicitly addressing this. Perhaps authors could consider revising to discuss whether they believe this to be biologically meaningful, or rather some artifact arising from something in the way the data was handled.
Results in general; Could you provide results tables with the full model outputs?
Figures; I think overall these are a nice way of showing the data, but it could also be nice to see the actual spread of the data (rather that summarising in boxes)

Reviewer 3 ·

Basic reporting

The MS was relatively well written with a good structure, good use of references and mostly clear and unambiguous writing. I did find minor mistakes in some of the that pervaded across the MS. These should be relatively easy to fix. I detail the specifics below in the general comments section.

Experimental design

The MS reports a relatively simple experiment in which the authors provided focal lizards with cotton wool buds containing the scents of different individuals and measured their response. This experimental design is fairly standard for studies of responses to olfactory cues and appeared to be relatively well executed. These experiments were placed in the appropriate context within both the introduction and the discussion.

Validity of the findings

The authors aim was to study kin recognition in a amphisbaenian reptile which has been suggested to live in stable social groups. Such simple social systems are excellent model systems for trying to understand broader questions about the origins and evolution of social organization linked to family life. Central to understanding this is understanding how key communication traits that mediate interactions between individuals (the building blocks of such social groups) go vary with aspects of the social living. The authors do a good job of placing their research in this context and site much of the relevant literature related to this topic. The experimental design is relatively simple but appears well executed. The raw data provided looks robust with respect to the recording of tongue flick rates. I had a few queries about the ways in which the data were analyzed which I detail in the next section. The conclusions appear well stated and backed up and the authors did a good job of not taking their results too far given that lack of paternity assignment and thus knowledge of the actual relatedness between individuals (e.g., the extent to which the authors could infer kin recognition).

Additional comments

I had a number of suggestions with respect to the MS that I hope improve the strength of the paper. I detail these below.

Title: In line with your inability to assign or know parentage between the offspring and adults – should this be altered to “Offspring and Adult Chemosensory recognition…”?

Lines 47 – 48: This sentence jumps out a little – I wonder whether you need it. The link to viviparity are important in some respects (see below) but I am not sure it is important to mention up here. Furthermore, you use both viviparous and oviparous examples in the rest of the paragraph.

Lines 56 – 58: I did not get a feel for the big sell of this paper here in terms of our understanding of co-variation between social evolution and recognition. Just a sentence or two pointing out perhaps that an increased understanding of the diversity of social life in reptiles prompts deeper questions about the mechanisms by which they are maintained (one mechanism by communication).

Lines 72 – 74: It might be worth pointing out that these simple suggested forms of parent-offspring associations are similar to those identified for other lizard species, including those that live in stable family groups (e.g., Gardner et al. 2016; Halliwell et al. 2017).

Lines 94: The formation and maintenance of family groups?

Lines 127: Given you don’t know these are family groups – is there another way to articulate this – “Individuals who were found together in groups were maintained together in separate”

Line 135: Should this be thigmothermy?

Lines 202 – 203: It is not quite clear what you mean by each scent stimulus was used once – does this mean that once you used one stimulus (e.g., cotton wool bud) it was thrown away and a new one used for the next focal animal? I also assume this means that each individual was tested on subsequent days for the rest of the scent combinations – so an adult was tested 3 times and a juvenile 5 times.

Lines 211 – 214: Given that the same individual was included in the data set multiple times (e.g., each juvenile was tested five times and each adult three times) I think you might need to control for subject id in your models with a mixed modelling approach.

Discussion general: I wonder whether you can go back to some of the original points of the introduction and the links to viviparity to explain some of your results. For example, you found that females responded the same to all offspring whereas males responded more strongly to familiar conspecifics. To me this makes sense, given that females will always be present at the production of their offspring whereas males will not necessarily. This might mean that there is limited “need” for females to recognise their own offspring during these early stages of parent-offspring association – because they are always there. In contrast, the male may not be present and therefore there may have been stronger selection on males to distinguish between familiar and unfamiliar offspring. This of course requires there to be some costs of associating with offspring – but that could simply be the costs associated with not eating them if normally there is conspecific infanticide.

The same arguments could be used to explain the juvenile results. As with above, offspring are always going to be associated with their mother upon birth. Therefore, if individuals do not disperse from their natal site, they have no need to necessarily recognise their mother. The adult male however may not be their father – and thus there may be stronger selection on recognising between their own dad and another male.

This would explain both sets of results with the same mechanism. Given the points raised about viviparity and the role it plays in mediating social evolution in these taxa – this may represent a good opportunity to discuss some of the broader implications of viviparity for mediating selection on recognition mechanisms.

Lines 266 – 269: More broadly these simple recognition mechanisms may be the first steps towards more complex forms of parental care – and more complex forms of family life. This is probably a point worth making somewhere here.

Lines 271: change to “that may reduce infanticide by other individuals, such that offspring”

Lines 300 – 302: And this point is more relevant to females than males – that is they should always be found with their mothers (if their mothers give birth to them) but not necessarily with their fathers (depending on the mating season and levels of dispersal).

Lines 305 – 308: Is there a precedence for where different entities within a species differ in the variation of their chemical signals in a way that mediates the response?

List of the minor corrections I identified (this may not be exhaustive).

Line 61: change to “make these studies difficult”?

Line 63: change to “spend”

Line 68: Not sure this sentence needs “Moreover”

Line 71: changed to “maintained for several months into the next season”

Line 77: remove “)”

Line 90: change to “can discriminate”

Line 92: change to “have self-recognition abilities” or something similar.

Line 115: change to “lifted”

Line 127: change to “separate”

Line 130: change to “by keeping individuals together”

Line 142: change to “within”

Line 164: change to “display juvenile recognition”

Line 184: change to “undertaken”

Line 198: change to “had the ability to detect conspecifics using chemosensory communication”

Lines 202 – 203: How often were individuals swabbed for their scent during this time – did this overlap
with the actual tongue flick assays?

Line 244: remove “.”

Line 286: remove “On the other hand,”

---

## Round 0.2 · Minor Revisions

I think the manuscript is looking really good, and as I have said before I really think this is a fascinating and useful study! I have just gone over the manuscript myself and have made some more suggestions for improving the wording in a few locations. I also found a number of typos that need to be corrected (just little things like double-spacing, incorrect punctuation, and small things like that). I think I caught most of the typos, but in this last stage please take your time and carefully go over the entire manuscript to ensure no other typos exist.

Have a look at the attached word doc file with your and my tracked changes together. I think all you need to do is go through it, make the changes, and submit a cleaned and ready to submit a completed manuscript (i.e., with no tracked changes). After that, I think this manuscript should certain be ready for acceptance!!!

p.s. The system here will only let me upload a PDF file (see attached), but I will directly email you the .doc file with the track changes so it is easier for you to make them easily, quickly, and completely. Sorry about the technical issue.

---

## Round 0.3 · accepted · Accept

Thank you for the super quick turn around time! I am looking forward to seeing this manuscript in 'print' and seeing its impact in the field of reptile sociality moving forward.